# The Role of BMP Signaling in Osteoclast Regulation

**DOI:** 10.3390/jdb9030024

**Published:** 2021-06-28

**Authors:** Brian Heubel, Anja Nohe

**Affiliations:** Department of Biological Sciences, University of Delaware, Newark, DE 19716, USA

**Keywords:** osteoclast, BMP, osteoporosis

## Abstract

The osteogenic effects of Bone Morphogenetic Proteins (BMPs) were delineated in 1965 when Urist et al. showed that BMPs could induce ectopic bone formation. In subsequent decades, the effects of BMPs on bone formation and maintenance were established. BMPs induce proliferation in osteoprogenitor cells and increase mineralization activity in osteoblasts. The role of BMPs in bone homeostasis and repair led to the approval of BMP2 by the Federal Drug Administration (FDA) for anterior lumbar interbody fusion (ALIF) to increase the bone formation in the treated area. However, the use of BMP2 for treatment of degenerative bone diseases such as osteoporosis is still uncertain as patients treated with BMP2 results in the stimulation of not only osteoblast mineralization, but also osteoclast absorption, leading to early bone graft subsidence. The increase in absorption activity is the result of direct stimulation of osteoclasts by BMP2 working synergistically with the RANK signaling pathway. The dual effect of BMPs on bone resorption and mineralization highlights the essential role of BMP-signaling in bone homeostasis, making it a putative therapeutic target for diseases like osteoporosis. Before the BMP pathway can be utilized in the treatment of osteoporosis a better understanding of how BMP-signaling regulates osteoclasts must be established.

## 1. Role of BMP in Bone Formation

Bone is formed in two major ways: endochondral ossification and intramembranous ossification. The majority of the human skeleton is formed by endochondral ossification, excluding the flat bones of the skull, the mandible, and the clavicle [1]. Endochondral ossification begins during week six and seven of embryonic development with the differentiation and condensation of mesenchymal stem cells into chondrocytes [2]. Chondrocytes then lay down a framework for skeletal elements that is later mineralized by osteoblasts and remodeled by osteoclasts (Figure 1A,B). The initial mineralization takes place in the primary ossification center near the middle region of the skeletal element, known as the diaphysis (Figure 1C). The mineralization and elongation of the skeletal element remains active after birth until 25 years of age [2].

Each stage of endochondral bone formation is regulated by BMPs, including the activity of osteoclasts. In the initial stages of bone formation, BMP2 and BMP7 stimulate the proliferation and differentiation of mesenchymal stem cells into chondrocytes [3,4]. Later, the condensation process of chondrocytes is regulated by BMP-signaling transmitted through the phosphorylation and activation of SMADs [5,6]. Then, BMP-ERK1/2-signaling induces the proliferation and maturation of the chondrocyte condensation, encouraging these cells to grow into a proper framework for eventual bone formation through the regulation of Indian hedgehog (IHH) and parathyroid hormone (PTH) expression gradients [5,7,8]. Disruption in BMP-signaling during chondrocyte condensation results in deformed cartilage growth zones and short limb phenotypes [9].

After the framework of chondrocytes is formed, blood vessels invade the cartilage through a process called angiogenesis (Figure 1C). BMP-signaling further regulates this process, as the absence of the *BMP receptor type 1a* (*BMPRIa*) gene in vivo leads to impaired angiogenesis [10]. Once the blood vessels have penetrated the established cartilage, they bring in mature osteoblasts and osteoclasts, thus beginning the mineralization process [11,12,13]. Additionally, up to 60% of mature chondrocytes also transdifferentiate into osteoblasts themselves, adding to the mineralization of the extracellular matrix surrounding chondrocytes [14]. The transdifferentiation of chondrocytes is regulated by essential factor, Runt-related transcription factor 2 (RUNX2), that acts down stream of BMP-signaling [15]. Without proper of BMP-signaling chondrocyte transdifferentiation is lost in *BMPRIa* knockout mice [16]. 

While the mineralization process takes place, osteoclasts work to remodel the inner bone tissue to extend a network of blood vessels. (Figure 1D). The network of blood vessels within bone tissue is so extensive that it receives around 10% of the total cardiac output [17,18]. Due to this circulation of blood containing osteoclasts and osteoblasts, bone is capable of remodeling its structure and repairing damage, unlike avascular cartilage [19]. Once the primary ossification center has formed, secondary ossification centers develop at either end of long bones within the epiphysial plates, effectively shutting down the growth zones by age 25, preventing further elongation of skeletal elements [2,20]. The first remodeling process occurs at these ossification centers as osteoclasts resorb bone in the diaphysis. As a result, osteoclasts form a space for bone marrow to be stored, known as the medullary cavity (Figure 1E) [21]. The administration of BMP2 into the medullary cavity results in an initial increase in bone formation that is later lost [22]. The loss of bone density is attributed to the increase in absorption activity of osteoclast, indicating a dual role of BMP2 in bone homeostasis. With the emerging role of BMP-signaling in osteoclast differentiation, the same focus given to osteoprogenitors should be given to osteoclast progenitors. This is highlighted by the BMP-dependent ERK1/2 signaling in both osteoprogenitors and osteoclast progenitors. 

Furthermore, it is has been established that ERK1/2 phosphorylation leads to an increase in mesenchymal stem cell proliferation [23,24,25]. However, activation of ERK1/2 also leads to an increase in proliferation of preosteoclasts [26,27,28]. Regulation of osteoclast differentiation then helps bone remolding at the secondary ossification centers. Osteoclast resorption at the epiphysis creates a complex microstructure important for support and shock absorption.

## 2. The Role of BMP and Osteoclasts in Maintaining Bone Microstructure

To avoid fractures, bone cells must maintain a healthy density while sustaining a complex supporting microstructure. That includes the trabecular network of spongy bone and surrounding cortical bone (Figure 2). To sustain a healthy bone structure, the mature skeleton is constantly remodeling, breaking down old worn-out bone and producing new bone in its place. The continued activity of osteoblasts and osteoclasts results in the complete turnover of an adult skeleton roughly every ten years [29]. This turnover is important, allowing for the release of growth factors, like BMPs, as well as calcium and phosphorus trapped in the mineralized matrix [30,31,32]. BMP-signaling helps maintain this healthy bone density by stimulating osteoblast mineralization, differentiation and survival [33]. However, it has become clear that BMPs not only aid in the anabolic process of bone turnover but also the catabolic processes [34]. To that end, the loss of the BMP receptor type 1 a (BMPRIa) increased trabecular bone thickness due to a decrease in osteoblast activity and a larger decrease in osteoclast activity [35]. Without proper BMP-signaling, the imbalance between osteoblast and osteoclasts results in reduced bone quality.

Healthy bone requires a suitable microstructure, organized to support weight and absorb shock. Beyond the formation of the medullary cavity, osteoclasts play a vital role in creating and retaining important porous structures in long bones (Figure 2) [21]. These structures make up trabecular bone, a weaving network of pores filled with red bone marrow and adipose tissue. The storage of marrow here is essential as it contains hematopoietic stem cells that give rise to many cell types, including red blood cells, immune cells, and osteoclasts [36]. Remarkably, the complex arrangement of trabecular bone that surrounds these pores provides structural integrity to withstand and transfer shock from falls and daily strain [37,38]. This means that the number of trabecular connections, as well as their thickness, needs to be tightly regulated to allow for the storage of marrow as well as shock absorption. 

Surrounding the trabecular network is cortical bone or, the bone collar. Upon impact, trabecular bone transfers shock to the cortical bone, which is thick yet flexible, allowing for further shock absorption. To maintain proper trabecular connections and cortical thickness, osteoblasts create new bone and osteoclasts, in turn, resorb old bone (Figure 2). The balance between osteoclast and osteoblasts is regulated by BMPs. As a result, the loss of osteoblastic BMP2 causes a decrease in cortical bone thickness resulting in weaker long bones [39]. Beyond this, the effect of altered BMP-signaling in osteoblasts and the corresponding change in bone quality is well established [40,41,42]. While the effect of altered BMP-signaling in osteoclast differentiation and their impact on bone microstructure is underappreciated.

## 3. The Role of BMP in Stages of Osteoclast Differentiation

Originating from hematopoietic stem cells within the bone marrow, osteoclasts go through several stages of differentiation before becoming mature and active [43,44]. First, CD34+ hematopoietic stem cells differentiate into CD14+ monocytes [45]. Then, the monocytes differentiate into mononuclear preosteoclasts, which fuse to form polykarons or immature/inactive osteoclasts that are CD14- [46,47]. Lastly, the multinucleated polykaryons become active osteoclasts producing acid to break down calcified bone in absorption pits under their ruffled borders called Howship lacunae (Figure 3). These stages of differentiation are driven by a few critical factors: Macrophage Colony Stimulating Factor (M-CSF), Receptor Activator Nuclear Factor kb Ligand (RANKL), and BMPs. The roles of both RANKL and M-CSF in osteoclast differentiation were well studied, while the requirement of BMP-signaling is underappreciated [48,49,50,51,52]. Recent studies on the inhibition of BMP-signaling confirm that BMPs play a major role in RANKL-mediated osteoclast differentiation, proliferation, fusion, and survival [36,53,54,55,56,57,58,59,60,61]. The addition of M-CSF begins the initial differentiation of hematopoietic stem cells. M-CSF upregulates PU.1, a transcription factor, which in turn induces the expression of Receptor Activator of Nuclear Factor kappa-B (RANK) on the membrane of osteoclast precursors (Figure 3) [62,63]. In addition to M-CSF, BMP4 also helps to upregulate PU.1 transcription initiating differentiation [64]. A combined signaling pathway between M-CSF and RANK then upregulates genes required for preosteoclast fusion, including the master fusion regulators *DC-STAMP*, *OC-STAMP,* and *ATP6v0d2* [65,66,67]. The expression of *DC-STAMP* and *ATP6v02* are regulated by an essential transcription factor, NFATc1, which is activated downstream of RANKL and BMP-signaling [68,69]. BMP inhibition leads to a decrease in *DC-STAMP*, *ATP6v02,* and *NFATc1* gene expression [55,61,70,71,72]. As a result, fewer, smaller, and less active osteoclasts form, showing the requirement of BMP-signaling in preosteoclast fusion [55,57,59,72]. Furthermore, BMP2 and NFATc1 are also required in immature osteoclasts for the proper induction of osteoclast specific genes as BMP2 promotes the nuclear translocation of NFATc1, which in turn upregulates the expression of TRAP and CATH K [69,73,74,75,76,77,78]. It is currently thought that BMPs only enhance RANKL mediated osteoclastogenesis. However, a study utilizing a soluble form of BMPRIa in bone marrow macrophages showed that in the absence of BMP-signaling, RANKL alone was not efficient in generating osteoclasts. Therefore, BMP-signaling may be required for RANKL-mediated osteoclastogenesis [54].

Several inhibitors are also known to regulate RANKL and BMP-driven osteoclastogenesis [55,61,72,79,80,81,82]. The most well-known inhibitor, osteoprotegerin (OPG), inhibits RANKL by binding to the ligand, preventing RANK-signaling. As such, an increase in OPG concentration may cancel out any increase in RANKL production. Like OPG, BMP inhibitors Twisted gastrulation 1 (TWSG1) and Noggin reduce BMP-mediated osteoclastogenesis [55,61,71,72]. This points to a more significant role for BMPs in osteoclast regulation and communication. 

## 4. The Communication of Osteoclasts with Bone and Immune Cells

The catabolic activity of osteoclasts and the anabolic activity of osteoblasts are tightly regulated to ensure a healthy bone structure. For instance, in the bone disease, osteopetrosis, too little resorption reduces bone turnover, causing an accumulation of older more fragile bone [83]. While in the bone disease, osteoporosis, too much bone resorption leads to a decrease in bone density and higher risk of fractures [84]. To understand how osteoclast activity is regulated, the communication of immune cells and other bone cells with osteoclasts must be understood. The RANKL:OPG ratio is the most well-researched avenue utilized to regulate osteoclasts [85,86,87,88,89]. Many cell types, including immune cells, manipulate the RANKL:OPG ratio by stimulating osteoclast differentiation and or increasing their activity [90,91,92]. RANKL is an essential ligand in driving osteoclastogenesis that is inhibited when bound to OPG [81]. By producing RANKL, immune cells directly regulate the RANKL:OPG ratio and increase osteoclast activity [93]. In addition to this a growing body of research, immune cells also regulate osteoclasts through the release of inflammatory cytokines, both directly and indirectly, by altering the RANKL production in osteoblasts (Figure 4) [42,94,95,96]. As such, proinflammatory cytokines such as Tumor Necrosis Factor-alpha (TNF-α), Interleukin 6 (IL-6), Interleukin 23 (IL-23), and interleukin 1 beta (IL-1β) released by immune cells, including macrophages and dendritic cells, are major contributors to osteoclast activation and differentiation [64,83,94,95,97,98,99,100,101]. Immune cells also inhibit osteoclasts through the release of cytokines such as Interleukin 4 (IL-4), Interleukin 18 (IL-18), Interleukin 33 (IL-33) and Interferon (IFN) [99,100,101,102,103,104]. The impact of these inflammatory cytokines is crucial in regulating osteoclasts after a fracture [105].

Bone cells such as osteocytes and osteoblasts also communicate with osteoclasts to modulate their activity and differentiation. Osteocytes, which are mature osteoblasts that have become embedded within bone, signal osteoclasts through direct cell-to-cell contact. Osteocytes do so by extending a dendritic process with membrane-bound RANKL to the bone surface to interact with membrane bound RANK on osteoblasts (Figure 4) [106,107]. Osteocytes also secrete soluble RANKL to stimulate osteoclasts, but to a lesser extent (Figure 4) [107]. Additionally, osteoblasts communicate with osteoclasts through paracrine signaling or gap junctions, similar to osteocytes [80,108]. Whether osteocytes or osteoblasts are the main sources of RANKL for the regulation of osteoclast activity and differentiation is not yet established [109,110]. Nevertheless, both cell types play a critical role in communicating with osteoclasts to maintain bone homeostasis. The crosstalk between immune cells, osteoblasts and osteoclasts creates a network of communication. Ultimately resulting in the regulation of osteoclast differentiation and activity required to maintain bone homeostasis. 

### Indirect Effect of BMP-Signaling on Osteoclast Communication

Osteoblasts and osteoclasts are in constant communication, recruiting and stimulating each other to maintain proper bone homeostasis and repair bone damage. They do so by sending cytokines via paracrine signaling and contact with membrane-bound ligands [60,106,107]. The roles of RANKL, OPG, and inflammatory cytokines were well studied in their communication with osteoclasts [111,112,113]. However, the importance of BMP-signaling in the communication and regulation between osteoclasts and osteoblasts is underappreciated. Osteoclasts lacking *BMPRIa* caused an increase in osteoblast activity by increasing gap junction communication with osteoblasts [60]. Inversely, osteoblasts lacking *BMPRIa* caused a reduction in osteoclastogenesis due to a subsequent reduction in sclerostin, a glycoprotein that stimulates the release of RANKL in osteocytes [114,115,116]. The release of BMP6 by osteoclasts also recruits and stimulates osteoblasts, promoting mineralization [117,118,119,120]. Furthermore, a reservoir of growth factors including BMP2, BMP4, and BMP 7 are stored in bone tissue. Osteoclasts release these stored growth factors via bone resorption [121,122,123]. The release of these factors thereby stimulates osteoblasts. Stimulated osteoblasts in turn release factors that regulate osteoclast differentiation and function, including M-CSF, which is required to drive hematopoietic stem cells into the osteoclast lineage [76,124]. Furthermore, BMP2 and 4 alter the RANKL:OPG ratio by modifying the levels of either RANKL or OPG released from osteoblasts [54,70,86,88]. The conditional knockout of *BMPRIa* in osteoblasts decreased the RANKL:OPG ratio [114]. This establishes the indirect effect of BMP-signaling in regulating the RANKL:OPG ratio. The recruitment of osteoblasts to a fracture site highlights the significance of this communication. When a fracture occurs, osteoclasts come in to resorb the damaged bone and then recruit osteoblasts to lay down new bone at the site of injury through the release of growth factors including, BMP2, BMP4, BMP6 and 7 [69,121,125]. In the absence of BMP2 fracture healing does not begin, highlighting the requirement for its release during bone resorption [126]. Beyond indirect effects of BMPs, a growing body of studies indicates that BMPs have more than just an indirect effect on osteoclasts [53,54,55,56].

## 5. Direct Effect of BMP-Signaling in Osteoclasts

First, it was determined that purified osteoclasts and osteoclast precursors express BMP receptors. Then, it was established that BMP2 directly stimulates osteoclast precursors with the help of M-CSF by Kanatani et al., 1995, thus suggesting a greater role for BMPs in bone homeostasis. This was also supported by the increased expression of the preosteoclast fusion protein, DC-STAMP, upon BMP2 stimulation [55]. In addition to BMP2, BMP4 and BMP6 also stimulate osteoclasts, but to a lesser degree [56,57,59,127]. To elucidate the potential direct effect of BMPs on osteoclasts, the BMP pathway was disrupted by Twisted gastrulation 1 (TWSG1) and Noggin, a BMP antagonist, resulting in the inhibition of osteoclast differentiation [55,61,70,71,72]. Later, the conditional knockout of the BMPRIa receptor resulted in a decrease in osteoclastogenesis and *DC-STAMP* expression [58]. Furthermore, the deletion of *BMPRIa* in vivo led to an increase in trabecular bone and bone density, along with a decrease in osteoclast activity and osteoclast differentiation [35,58,128]. A deletion of *BMPRIb* also showed an increase in proliferation of preosteoclasts and increase differentiation while decreasing resorption [129]. Additionally, the deletion of *BMPR2* decreased osteoclast differentiation and activity [53]. Taken together, BMP-signaling plays a critical role in not only osteoblasts but also the proper differentiation of osteoclasts, making it an ideal target for future therapeutic treatments.

## 6. BMP-Signaling Pathway in Osteoclasts

BMPs signal either through the canonical or noncanonical pathways. The canonical pathway is also known as the SMAD signaling pathway. The SMAD pathway involves a few different types of SMADs including R-SMADs (SMAD1/5/8), which are receptor-regulated SMADs, and Co-SMADs such as SMAD 4 that bind to R-SMADs, facilitating nuclear translocation for signal transduction (Figure 5) [130]. Multiple studies established that osteoclasts express SMADs as well as phospho-active SMADs [53,54,55,131,132]. Furthermore, research suggests that BMP-mediated SMAD signaling may play a role in osteoclast fusion and activation, as inhibition of SMAD leads to smaller and less active osteoclasts [53,69,132,133]. However, SMAD signaling is unaffected in *BMPRII* knockout in osteoclasts, suggesting that BMPs primarily signal through the noncanonical pathway in osteoclasts [53,131]. The noncanonical signaling pathway consists of mitogen-activated protein kinase MAPK downstream signaling molecules including c-Jun N-terminal kinase (JNK), mitogen-activated protein kinase 38 alpha (p38α), and extracellular regulated kinases (ERK), all of which are activated by BMP2 in osteoclasts [53,131]. TGF-β Activated Kinase 1 (TAK1), an upstream signaling molecule of the noncanonical signaling pathway, is required for osteoclast differentiation. The specific knockout of *TAK1* in osteoclasts leads to an osteopetrosis-like phenotype with decreased resorptive activity [134]. Furthermore, while not shown in osteoclasts, X-linked inhibitor of apoptosis protein (XIAP) associates with both BMPRIa and TGF-β Activated Kinase 1 binding protein (TAB1) [135]. TAB1 then forms a complex with TGF-β Activated Kinase 1 binding protein 2 (TAB2) associating with and activating TAK1. This indicates that the BMP/BMPR-XIAP-TAB1/TAB2-TAK1 pathway allows for BMPs to stimulate the noncanonical pathway including MAPK’s JNK, p38, and ERK in osteoclasts. Further, this overlaps with the RANK signal transduction, which also uses the MAPK pathway, activating downstream the molecules JNK, p38, and ERK [27,136]. The difference in RANK signaling is that Tumor Necrosis Factor Receptor-associated Factor Protein 6 (TRAF6) rather than XIAP recruits TAB1/TAB2 to the RANK receptor, activating TAK1. MAPK signaling converges to increase the expression and translocation of NFATc1a, an essential transcription factor that leads to the upregulation of osteoclast genes for fusion and activation [69,73,137,138,139,140] This overlap in MAPK signaling between the RANKL pathway and the BMP pathway may contribute to the enhancement of osteoclastogenesis (Figure 5). 

## 7. The Role of Osteoclasts in Bone Disease

Diseases impact the health of bone by altering the microstructure and overall bone mineral density. These diseases primarily alter the function and balance of osteoblasts and osteoclasts. Two major diseases that impact bone density are osteopetrosis and osteoporosis. These diseases are two sides of the same coin, as they both impact osteoblasts and osteoclasts inversely. Osteopetrosis occurs when either osteoblast activity is increased or osteoclast activity is decreased, causing an imbalance in bone maintenance and an increase in bone mineral density. Osteopetrosis causes an increased risk of fracture as the trabecular network is disrupted [141]. Further ossification also leads to the shrinking of the medullary cavity reducing stem cell storage [142]. This leads to immune disruption and inhibition of healing [143]. Conversely, in osteoporosis, osteoblast activity is decreased, and osteoclast activity is increased. This imbalance favors bone resorption, causing a decrease in bone mineral density. The trabecular number and thickness decreases within spongy bone, resulting in a high risk of fracture [84,144,145]. Osteoporosis is one of the most prevalent bone diseases in the world, affecting 37% of women and 20% of men over the age of 50 [146]. The most common cause of osteoporosis is the onset of menopause, brought about primarily by a drop in estrogen in women [21,147]. The decrease in estrogen expression leads to an increase in the RANKL:OPG ratio osteoclastogenesis, promoting bone resorption [148]. Additionally, the onset of menopause decreases the BMP antagonist, inhibin A, stimulating both osteoblastogenesis and osteoclastogenesis. This suggests a dual role for BMP-signaling in bone metabolism of post-menopausal women [149]. 

As the aging population increases, a higher proportion of people are affected by osteoporosis for a greater portion of their lives [150,151]. The longer an individual suffers from osteoporosis, the more likely they are to experience fractures. Over 50% of the people that experience osteoporotic fractures cannot continue to live independently and more than 25% will die within 12 months after their injury [152,153,154]. Furthermore, the International Osteoporosis Foundation found that the risk of fracture from osteoporosis is such that one in every three women and one in every five men across the globe will experience an osteoporotic-related fracture. This results in a substantial annual medical cost with an estimate of 16 billion dollars spent in the United States in 2002 alone [155]. Moreover, the European Union spent more than 37 billion euros on osteoporosis-related treatment in 2010 [156]. Clearly, an effective long-term treatment for this disease is desperately needed worldwide.

## 8. Treatments for Osteoporosis

One of the early treatments developed for osteoporosis was hormone replacement therapy [157,158]. As the most common cause of osteoporosis is the post-menopausal loss of estrogen, the hope for this treatment was that, by replacing the lost hormones, they could restore homeostasis of bone regulation. Unfortunately, this treatment showed an increased risk of heart attacks, breast cancer, and strokes, causing it to discontinue [159,160]. The use of BMP2 has also been considered for treatment of osteoporosis due to its osteogenic effects post ALIF surgery [161,162,163,164]. However, increased resorption activity has been associated with this procedure, [165,166]. A total of 12 months after treatment, ALIF patients showed increased graft subsidence over patients that did not receive treatment with BMP 2 [166]. Additionally, treatment of osteoblasts isolated from patients with osteoporosis with BMP 2 showed no osteogenic response compared to osteoblasts derived from osteoarthritic patients [167]. As a result, the use of BMP2 for the treatment of osteoporosis is still under consideration. However, the dual impact on bone formation and bone resorption makes the BMP pathway a potential therapeutic target for osteoporosis.

Currently, the most popular treatment for osteoporosis is bisphosphonates. Bisphosphonates increase bone mineral density by inhibiting the mevalonate pathway, inducing apoptosis, and reducing osteoclast resorption [168,169,170,171]. Despite their popularity, they come with various side-effects, such as the increased risk of esophagus cancer, intestinal irritation, and osteonecrosis of the jaw [172,173,174]. Doctors may recommend taking a break from bisphosphonates after five years to reduce the risk of these side effects. Another antiresorptive drug called Denosumab is an alternative for bisphosphonates, acting as a RANKL inhibitor [175]. Denosumab blocks RANK signaling by binding to its ligand and preventing it from binding to the receptor on osteoclasts; therefore, reducing resorptive activity and increasing bone mineral density [175,176]. In opposition to antiresorptive drugs, other treatments have sought to increase the anabolic process of bone homeostasis by stimulating osteoblasts. One such new anabolic peptide, called p-PTH 1-34 (Teriparatide), stimulates osteoblast activity to increase bone mineralization, was recently approved for the treatment of osteoporosis [177,178,179]. This is a peptide derived from PTH, a known anabolic regulator of bone formation [180,181]. However, Teriparatide increased the risk of hypercalcemia [180,182]. Furthermore, long-term treatment causes an increase in RANKL production within osteoblasts, causing an indirect increase in bone resorption by osteoclasts [177].

Two potential therapeutics that have utilized the dual anabolic and antiresorptive properties include a sclerostin antibody, namely Romosozumab, and a Casein Kinase 2 (CK2), inhibitor (CK2.3). Sclerostin is a known inhibitor of osteoblast activity, released by osteoclasts to reduce anabolic activity [115,183,184]. Romosozumab is an antibody inhibiting sclerostin from binding to its Wnt receptor, thereby removing sclerostin’s negative effect on osteoblasts. Remarkably, Romosozumab not only stimulates osteoblasts, but it also inhibits osteoclasts [185]. This antiresorptive effect is likely due to increased OPG production, produced downstream of the Wnt pathway. However, OPG is not always increased after treatment with other sclerostin inhibitors, thus a separate mechanism may be responsible for the antiresorptive effect of Romosozumab [186]. Furthermore, the effect of sclerostin inhibitors is attenuated in osteoblasts, likely due to decreased proliferation in osteoprogenitor cells [187]. Furthermore, Romosozumab increased the risk of cardiac events and should be closely monitored for extended treatment [188]. Overall, Romosozumab is a potentially valuable alternative to bisphosphonates.

The use of BMP2 has also been considered for the treatment of osteoporosis due to its osteogenic effects post ALIF surgery [48,161,163]. However, increased resorption activity has been associated with this procedure, [165,166]. A year after BMP2 treatment ALIF patients showed increased graft subsidence over patients that did not receive surgery with BMP2 treatment [166]. Additionally, BMP2 stimulated osteoblasts isolated from patients with osteoporosis showed no osteogenic response when compared to osteoblasts derived from osteoarthritic patients [167]. As a result, the use of BMP 2 for the treatment of osteoporosis is still under investigation. Nevertheless, the dual impact on bone formation and bone resorption makes the BMP pathway a potential therapeutic target for osteoporosis. The peptide CK2.3 utilizes the BMP pathway in the absence of a BMP ligand to stimulate osteoblast mineralization and inhibit osteoclast resorption [140,189,190,191,192]. Furthermore, CK2.3 increases mineralization in osteoporotic patients where BMP2 is not [167]. While the effects of CK2.3 are still being analyzed, it offers a new approach to the use of BMP-signaling in the treatment of osteoporosis.

## 9. Conclusions and Future Directions

BMP-signaling is essential for proper bone formation and maintenance. BMPs regulate the condensation of chondrocytes in endochondral ossification as well as the transdifferentiation and activation of osteoblastic bone mineralization. Improper BMP-signaling results in short limb phenotypes, highlighting its importance during skeletal development. Mature bone is also regulated by BMP-signaling, as it maintains a balance of anabolic and catabolic activity between osteoblasts and osteoclasts. The requirement for BMP-signaling in osteoblasts has been established as a potent driver of bone mineralization. However, increasing evidence points to an additional role BMP-signaling in bone resorption. This is supported by an increase in trabecular bone and bone density in BMPR1a knockout models, attributed to a decrease in osteoblast activity and a more significant decrease in osteoclast differentiation and activity [35,58,128]. Despite this, the research on BMPs has primarily focused on osteoblasts.

The dual role of BMP-signaling in bone homeostasis presents it as a potential treatment for Osteoporosis. As osteoporosis is the result of increased resorption and decreased mineralization of bone, a treatment that can impact both osteoblasts and osteoclasts is imperative. However, the majority of currently available treatments focus on just one side of bone metabolism. This presents a problem for long-term treatment as osteoblasts and osteoclasts are in constant communication, regulating one another to maintain suitable bone mineral density. A problem that BMP2 treatment itself faces. Treatment with BMP2 in spinal fusion surgeries results in increased bone mineralization followed by an increase in osteoclast absorption. To utilize the dual effect of the BMP-signaling pathway a treatment called, CK2.3 was created. CK2.3 stimulates the BMP pathway in the absence of BMP2, resulting in the stimulation of osteoblast and inhibition of osteoclasts. However, more research is still needed to determine its effectiveness for the treatment of osteoporotic patients. With the emerging role of BMP-signaling in bone resorption, a greater focus should be given to its impact on osteoclast regulation.

## Figures and Tables

**Figure 1 jdb-09-00024-f001:**
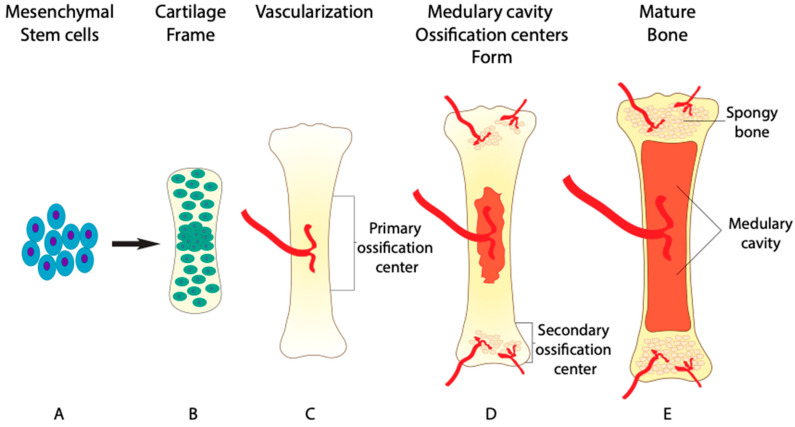
Endochondral bone formation. The phases of bone maturation and remodeling. (**A**) condensation of mesenchymal stem cells differentiating into chondrocytes. (**B**) Chondrocyte cells organize into a framework for the maturing skeletal element. (**C**) Initial vascularization beginning primary ossification at a long bone diaphysis. (**D**) Formation of secondary ossification centers and initiation of osteoclast-driven remodeling of medullary cavity and spongy bone. (**E**) Mature skeletal element with fully formed trabecular bone structure and medullary cavity filled with bone marrow.

**Figure 2 jdb-09-00024-f002:**
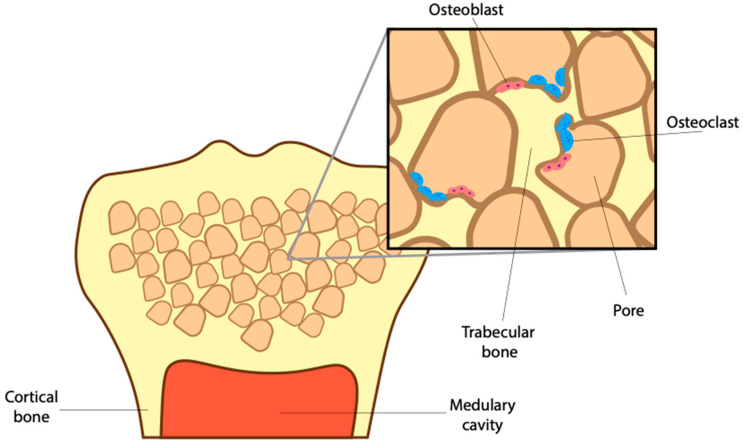
Maintenance of spongy bone microstructure. The epiphysial head of a long bone containing pores of adipocytes and stem cells (in beige). Surrounded by trabecular bone (in yellow). Osteoclasts (in blue) absorb bone and expand pores while osteoblasts (in pink) mineralize new bone closing pores and thickening trabecular bone. Cortical bone and medullary cavity run down to the diaphysis of the skeletal element (not shown in this image).

**Figure 3 jdb-09-00024-f003:**
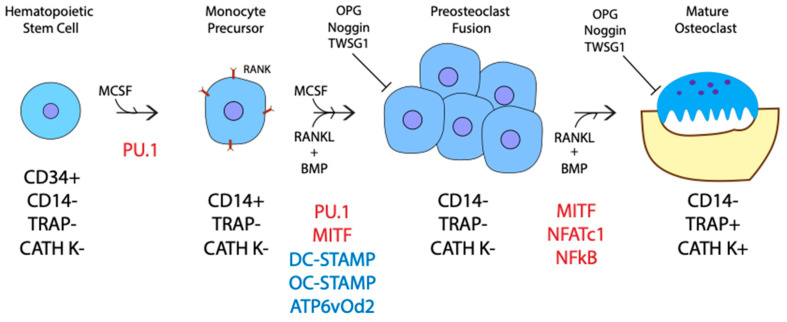
Osteoclast development. Hematopoietic precursor cells stimulated by mammalian colony stimulating factor (M-CSF) are driven toward the osteoclast lineage. M-CSF stimulates transcription factor PU.1 to express RANK on progenitor cells. CD14+ monocytes stimulated by RANKL, along with BMP4, help to upregulate the MITF and PU.1 transcription factors to increase expression of essential fusion proteins DC-STAMP, OC-STAMP and ATP6vOd2. Preosteoclasts fuse to form a polykaron. BMP2 and RANKL further the activation of mature osteoclasts and the transcription factors Microphthalmia-associated transcription factor (MITF), Nuclear factor of activated T-cells, cytoplasmic 1 (NFATc1) and nuclear factor kappa light chain enhancer of activated B cells (NFKB) upregulate osteoclast specific markers tartrate-resistant acid phosphatase (TRAP) and cathepsin k (CATH K). Mature osteoclasts form resorptive pits, or Howship lacunae, by pumping CATH K and TRAP against the bone surface, breaking down the mineralized bone. OPG, Noggin and Twisted gastrulation 1 (TWSG1) inhibit RANKL and BMP mediate osteoclast fusion and activation.

**Figure 4 jdb-09-00024-f004:**
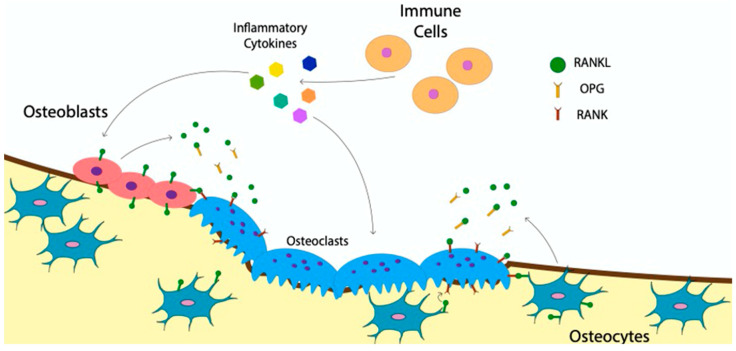
Osteoclasts communicate with immune cells and other bone cells. Osteoclasts communicate with immune cells through the releases of inflammatory cytokines. Osteoblast and osteocytes release soluble RANKL as well as osteoprotegerin (OPG) to communicate with osteoclasts. Osteocytes and osteoblasts also communicate with Osteoclasts via direct cell–cell contact utilizing membrane-bound RANKL.

**Figure 5 jdb-09-00024-f005:**
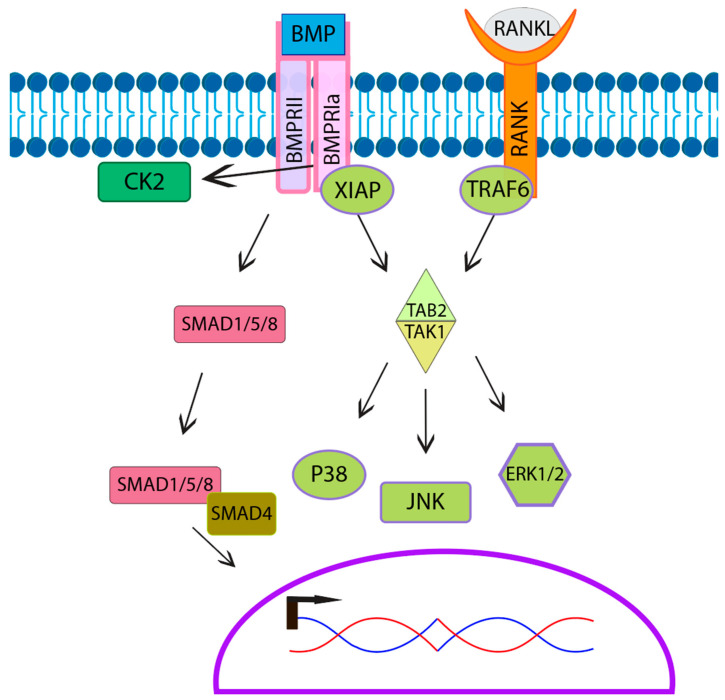
The BMP-signaling pathway and RANKL signaling overlap. BMP stimulates the downstream signaling pathways by activating BMP receptors BMPRIa and BMPRII. The canonical SMAD pathway transduces the signal into the nucleus with the help of co-SMADs (SMAD4). The noncanonical BMP-signal is transduced by the recruitment of TAB2/TAK1 complex the BMP receptor through XIAP. Thus, activating the same downstream effector proteins as the RANKL signaling pathway. These proteins can then modulate the expression of osteoclast genes.

## Data Availability

Not applicable.

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
