# Peer review of "The Role of BMP Signaling in Osteoclast Regulation"

_jdb, 2021, doi:10.3390/jdb9030024_

Round 1

Reviewer 1 Report

Overall, while this paper looks at most aspects of BMP signaling in the context of bone remodeling, I felt there was a strong emphasis on osteoclast functions and less so on osteoblast functions. The paper is overall well put together but would like to see a more equal emphasis on both catabolisms as well as anabolism. Many typos as well, and stylistically, need to make some of the molecule/receptor names in this paper consistent with what is commonly used in the literature. 

Reviewer 2 Report

This paper focuses on the role of BMP-signaling in osteoclasts mostly. This paper is not properly organized and does not satisfy the standard level of a review paper.  Followings are some critical points:

  • The abstract is quite misleading.  The authors do not fully and directly explain why BMP2 treatment for anterior lumbar interbody fusion raising concerns? What are the potential complications with long-term treatment exactly? What are the pros and cons? What is the alternative? If the focus of this paper is not on BMP treatment, then why the author raises this point in the abstract and never address the details in the body of the manuscript.
  • The authors explain Bone is formed in two major ways, endochondral ossification and intramembranous ossification. They further explain endochondral ossification in details. It would be better if they specify at what stage during fetal development. What about primary ossification after birth?
  • In the same section, Heubel et.al mention that chondrocytes transdifferentiate to osteoblasts. With what factor? What percentage? How it relates to the role of BMP signaling in bone formation? In the same section, authors claim that the role of BMP signaling in regards to osteoclast progenitor cells is under appreciated, but they never directly mention why this is important and what does it mean.
  • The numbering system correlated to the different sections of the manuscript is not correct. Starts with 1.1, 1.2 then it jumps to 1.5 then 1.4 etc.
  • There is some redundancy in the information.
  • In some sections, the authors provide the lists of events without linking them together in a comprehensive fashion to help the reader following the events.
  • The authors mostly focus on the BMP-signaling in osteoclasts, and overlook the role of the BMP signaling in osteoblast regulation. If the authors intend to mostly focus on the role of the BMP signaling in osteoclast regulation, then the title of the paper is misleading.
  • Considerable number of old published papers are cited in this work (there is really not much novelty here). Please consider reviewing recent papers.
  • Can the authors propose how the treatment for Osteoporosis is directly related to BMP signaling?
  • The authors claimed that: “the aging population increases; a higher proportion of people are affected by osteoporosis for a greater portion of their lives.” Is this a fact? Is there any validated statistics to support this claim?
  • The manuscript does not provide any conclusion and future direction.
  • Too many connectors and sometime informal language is used.

Round 2

Reviewer 2 Report

Thank you for addressing the issues.